# Understanding Computational Models of Semantic Change: New Insights from the Speech Community

**Filip Miletić**[1*]   **Anne Przewozny-Desriaux**[2]   **Ludovic Tanguy**[2]

[1]Institute for Natural Language Processing, University of Stuttgart

[2]CLLE, CNRS & University of Toulouse

`filip.miletic@ims.uni-stuttgart.de`

`{anne.przewozny, ludovic.tanguy}@univ-tlse2.fr`

## Abstract

We investigate the descriptive relevance of widely used semantic change models in linguistic descriptions of present-day speech communities. We focus on the sociolinguistic issue of contact-induced semantic shifts in Quebec English, and analyze 40 target words using type-level and token-level word embeddings, empirical linguistic properties, and – crucially – acceptability ratings and qualitative remarks by 15 speakers from Montreal. Our results confirm the overall relevance of the computational approaches, but also highlight practical issues and the complementary nature of different semantic change estimates. To our knowledge, this is the first study to substantively engage with the speech community being described using semantic change models.

## 1 Introduction

Research on computational analyses of semantic change has established a range of modeling approaches, standard evaluation practices, and evidence of their practical utility as well as shortcomings in descriptive linguistic applications (Tahmasebi et al., 2021). However, most work to date has focused on general-purpose diachronic corpora with little information on the underlying speakers, but the same methods are increasingly used to study finer-grained patterns in present-day speech communities. This raises the vital question of how the output of these models is perceived by the speakers whose linguistic behaviors they purport to describe.

We address this issue focusing on the sociolinguistic context of Quebec English, which exhibits cross-linguistic semantic change due to contact with French (Miletić et al., 2021). We analyze 40 semantic shifts in a regionally stratified corpus of tweets, obtaining semantic change estimates from type-level and token-level word embedding models,

as well as a range of empirical linguistic information. We use face-to-face sociolinguistic interviews with 15 speakers from Montreal to elicit acceptability ratings and qualitative remarks for the target semantic shifts attested in tweets from our corpus.

We provide the following contributions. (1) A quantitative comparison of model-derived semantic change scores, human acceptability ratings, and empirical linguistic properties, confirming the overall relevance of standard approaches and highlighting their shortcomings. (2) A qualitative analysis of the social values linked to semantic shifts, clarifying the complementarity of semantic change estimates. (3) A set of 40 target words with model-derived semantic change scores, empirical linguistic properties, as well as speaker-provided acceptability ratings, synonyms, and qualitative remarks.[1] To the best of our knowledge, this is the first study to substantively engage with a speech community in the process of describing its language use with semantic change models.

## 2 Related work

Semantic change over time is mainly analyzed using word embeddings (Kutuzov et al., 2018). Type-level models provide meaning representations that are specific to time periods and can be compared across them (Dubossarsky et al., 2019; Gulordava and Baroni, 2011; Hamilton et al., 2016; Kim et al., 2014). Token-level models provide contextualized embeddings that can be pooled into time-specific representations or used in clustering (Martinc et al., 2020; Giulianelli et al., 2020). Other deep neural architectures use cross-lingual transfer (Rachinskiy and Arefyev, 2022) or train on temporal data (Rosin and Radinsky, 2022). Similar methods are used to compare word meanings across online communities (Del Tredici and Fernández, 2017), text types

---

*Work done at the University of Toulouse.

[1]The resource is available at `http://redac.univ-tlse2.fr/misc/canenTestset.html`

(Fišer and Ljubešić, 2018), dialect regions (Kulkarni et al., 2016), and languages (Uban et al., 2019).

The prevalent evaluation approach uses graded type-level semantic change scores (Schlechtweg et al., 2018) to assess models on ranking or binary classification. This has enabled improvements in model design, shown by shared tasks on a range of European languages (Basile et al., 2020; Pivovarova and Kutuzov, 2021; Schlechtweg et al., 2020; Zamora-Reina et al., 2022). The models have been used in promising linguistic analyses (De Pascale, 2019; Rodda et al., 2019; Xu and Kemp, 2015), including of Quebec English (Miletić et al., 2021), but their descriptive value is yet to be fully determined (Boleda, 2020). Further validation can be provided by the targeted speakers, similarly to collecting semantic change ratings from online communities under study (Del Tredici et al., 2019). However, we argue for much more extensive engagement based on variationist sociolinguistic practice (Labov, 1972; Tagliamonte, 2006).

## 3 Data and method

We aim to gain a better understanding of previously proposed computational semantic change measures by examining how they relate to a standard sociolinguistic estimate of the same phenomenon – in this case, contact-induced semantic shifts in Quebec English. We begin by presenting the data and methods used to obtain these estimates.

### 3.1 Corpus of Canadian English tweets

We are unaware of any diachronic corpus of Quebec English, and therefore rely on synchronic data to model regional differences in meaning which indirectly reflect semantic change over time. We use a corpus of English-language tweets posted by users from Montreal, Toronto, and Vancouver (Miletić et al., 2020). We aim to find effects of language contact by detecting word usage specific to Montreal, the only of the three cities where French is widely used. We rely on the two control regions to limit the impact of unrelated regional variation.

The original corpus is filtered for language, location, and near-duplicate content. Like in previous work (Miletić et al., 2021), we only retain (i) tweets posted from 2016 onwards to limit spurious diachronic effects; (ii) users with at least 10 tweets in the corpus; (iii) a maximum of 1,000 tweets per user, with a random subsample where that is exceeded. The final corpus contains $\approx$ 35m tweets posted by $\approx$ 150k users, for a total of $\approx$ 630m tokens balanced across the regions.

### 3.2 Target semantic shifts

We use the only available dataset for semantic change detection in Quebec English, which we previously introduced (Miletić et al., 2021); it is similar in size to those from recent shared tasks (e.g. Schlechtweg et al., 2020). We retain 40 items corresponding to semantic shifts, which were identified in the sociolinguistic literature and through corpus exploration. The presence of contact meanings in our corpus was also manually validated. A typical semantic shift from the dataset is illustrated by the following tweets:

(1) This isn't the level of realistic **exposition** you'd require in fiction.

(2) I really want to go to an art museum or an art **exposition**.

In example (1), *exposition* is used with one of its conventional English meanings, 'introductory part of a narrative'. Example (2) illustrates the contact meaning 'art exhibition' – it is not generally used in English, but it is typical of the French homograph *exposition*. Its presence in Quebec English is likely explained by the locally widespread use of French.

We further extend the set of semantic shifts with several empirical linguistic properties shown to affect computational estimates of semantic change (Del Tredici et al., 2019; Hamilton et al., 2016; Uban et al., 2019). (1) **Frequency** is taken from the Montreal subcorpus. (2) **Polysemy** is calculated as the number of synsets in WordNet (Fellbaum, 1998). (3) We compute a target word's **regional specificity** to the Montreal subcorpus compared to the whole corpus using the Sparse Additive Generative model (SAGE) (Eisenstein et al., 2011). It is based on the maximum-likelihood criterion, with a regularization parameter ensuring that rare terms are not overemphasized. (4) We quantify a target word's **contextual diversity** by taking the mean pairwise cosine distance between the embeddings of its context words (from the type-level model described below). This is limited to 1,000 (possibly non-unique) contexts in a symmetrical 10-word window; they are randomly subsampled if needed.

### 3.3 Embedding-based models

We assess different types of semantic change estimates that can be derived across modeling strate-

gies. We use type-level models to obtain broad estimates, and token-level models to target individual contact-related occurrences. The setup is introduced below; see Appendix A for full details.

We learn **type-level word embeddings** using the best configuration from a previous evaluation on this dataset (Miletić et al., 2021) – word2vec models (Mikolov et al., 2013) trained separately for each subcorpus and aligned using Orthogonal Procrustes. For each word, we take the pairwise cosine distances for its vectors from models trained on the regional subcorpora, and quantify semantic change by averaging over the Montreal–Toronto and Montreal–Vancouver distances ($\mathbf{cos_{AVG}}$). We expect this score to capture the amplitude of regional semantic differences on the type level, and to be higher for clear-cut than for fine-grained semantic distinctions.

For each target word, we obtain **token-level word embeddings** of its occurrences using BERT-base-uncased (Devlin et al., 2019) and cluster them using affinity propagation, following Miletić et al. (2021). Up to 10 clusters with the highest proportion of tweets from Montreal are annotated by domain experts. They manually tag each cluster with a binary label indicating if the prevalent meaning in it is related to contact influence. For each target word, we calculate the proportion of tweets tagged as contact-related, out of all annotated tweets ($\mathbf{clust_{CONT}}$); and the proportion of tweets from Montreal, out of all tweets tagged as contact-related ($\mathbf{clust_{REG}}$). We expect $\mathbf{clust_{CONT}}$ to capture the diffusion of contact-related uses, as it implies a higher number of different usage contexts and users producing them. It should correlate with estimates of diffusion in face-to-face communication. By contrast, $\mathbf{clust_{REG}}$ measures the regional specificity of contact-related uses, which might vary for different reasons. For instance, an emergent semantic shift may be specific to Montreal, but an established one may spread elsewhere.

### 3.4 Speaker ratings and remarks

We conducted interviews with 15 Montrealers in early 2022.[2] We used the PAC-LVTI sociolinguistic interview protocol (Przewozny et al., 2020) to elicit detailed linguistic and sociodemographic background. This was followed by a semantic perception test in which they were shown our 40 target

semantic shifts; each was presented in one tweet from the corpus, jointly chosen by three annotators. We asked the participants to (i) read the tweet out loud; (ii) rate the acceptability of the target word from 1 to 6; (iii) provide a contextual synonym for the target word; (iv) provide any other comments. Given our use of face-to-face interviews to assess online communication, stylistic differences between the two situations may have a confounding effect. We aimed to limit it by prioritizing tweets which are not stylistically marked; asking the participants to rate the target word itself rather than properties of its context; and conducting the rating task at the end of the interview, when participants generally communicate more freely.

Acceptability ratings are widely used to quantify the perception of linguistic features in a speech community (Dollinger, 2015). Here, they reflect the local diffusion of contact-related uses: in direct terms, among our participants; indirectly, in the broader speech community, since more widespread general use of an item likely contributes to higher acceptability ratings. We compute mean acceptability ratings ($\mathbf{human_{AVG}}$) by averaging over all participants who interpreted the tweet with the posited contact-related meaning.[3] We include all words for which more than half of the participant ratings are retained (37 of 40); for these, 14.1 ratings are retained on average.

We transcribe the audio recordings of the semantic perception test so as to (i) extract the synonyms for the semantic shifts; (ii) annotate the associated social values: local specificity, for items described as typical of Quebec; French influence, for those described as typical of French speakers. At least one speaker expressed a social value for 26 semantic shifts. For the full interview protocol and speaker profiles, see Appendix B.

## 4 Results and discussion

### 4.1 Model-derived scores vs. speaker ratings

Speaker acceptability ratings show a weak negative correlation with the type-level score $\mathbf{cos_{AVG}}$ ($\rho = -0.370$, $p = 0.024$) and a trend towards a positive correlation with the token-level score $\mathbf{clust_{CONT}}$ ($\rho = 0.238$, $p = 0.155$). They are uncorrelated with $\mathbf{clust_{REG}}$ ($\rho = 0.091$, $p = 0.592$). While the two stronger relationships obtain inverse

---

[2]Future participants from Toronto and Vancouver may provide further relevant information, over and above this analysis.

[3]For items where all provided contextual synonyms correspond to the target interpretation, we also retain the ratings where no explicit synonym is given.

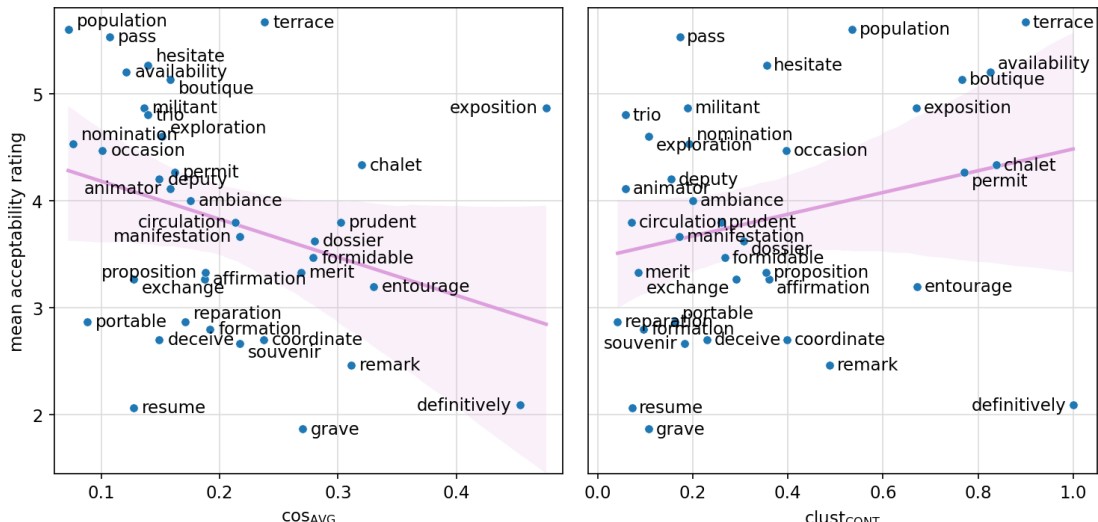

Figure 1: Regression plots comparing mean human acceptability ratings with model-derived semantic change estimates. Left: type-level cosine distance (average of Montreal–Toronto and Montreal–Vancouver); right: proportion of tweets manually annotated as contact-related following token-level clustering.

correlations due to their nature (semantic difference vs. diffusion), they confirm that model-derived estimates at least partly align with the target speakers' perception of semantic change. But the captured patterns are not the same, as shown in Figure 1.

The left panel suggests that semantic change is more accepted with a lower amplitude, as measured by type-level cosine distance. For example, *population* 'general public' (cf. Fr. *population*) has a high acceptability rating, but this is a slight shift from 'inhabitants'. Acceptability is much lower for *definitively* 'for sure' (cf. QF *définitivement*), but this is a stark change from 'conclusively'. Other trends point to technical issues: *exposition* 'exhibition' (cf. Fr. *exposition*) has a similarly high cosine distance but also a higher acceptability. Its French homograph is sometimes used in codeswitched tweets from Montreal, inflating its cosine distance.

The token-level score (right panel) measures the diffusion of the contact-related meaning in Montreal. It parallels acceptability ratings in some cases, such as *terrace* 'patio' (cf. Fr. *terrasse*). But other highly accepted semantic shifts like *trio* 'combo meal' (cf. QF *trio*) appear in few regionally-specific clusters, obfuscating their importance. By contrast, *definitively* 'for sure' would be described as widespread in the community based on corpus information, contrary to its low acceptability.

## 4.2   Effect of empirical properties

We further explore the differences between semantic change estimates by looking at their relation-

ships with empirical linguistic properties. Spearman's correlation coefficients are shown in Table 1; statistically significant results are discussed below.

|  | freq. | polys. | specif. | divers. |
|---|---|---|---|---|
| $\text{cos}_{\text{AVG}}$ | -0.764 | -0.378 | 0.450 | -0.439 |
| $\text{clust}_{\text{CONT}}$ | -0.083 | -0.371 | 0.146 | -0.093 |
| $\text{clust}_{\text{REG}}$ | -0.177 | -0.063 | 0.533 | -0.019 |
| $\text{human}_{\text{AVG}}$ | 0.396 | 0.096 | -0.091 | 0.254 |

Table 1: Correlation (Spearman's $\rho$) of semantic change estimates and empirical properties. Shading: $p < 0.05$.

**Frequency** shows a strong negative correlation with $\text{cos}_{\text{AVG}}$, similarly to monolingual (Hamilton et al., 2016) and contrary to cross-lingual (Uban et al., 2019) diachronic models. The link between frequency and cosine may reflect model artifacts (Dubossarsky et al., 2017), but we also find a moderate correlation with $\text{human}_{\text{AVG}}$, suggesting that frequency does play a role. **Polysemy** has a moderate negative correlation with both $\text{cos}_{\text{AVG}}$ and $\text{clust}_{\text{CONT}}$, contrary to diachronic studies (Hamilton et al., 2016; Uban et al., 2019). This might be due to the effect of more complex semantic properties, such as cultural relevance of lexical fields, on bilinguals (Zenner et al., 2012). **Regional specificity** shows a moderate positive correlation with $\text{cos}_{\text{AVG}}$ and $\text{clust}_{\text{REG}}$. This is consistent with regional semantic changes being paralleled by higher frequency in that region. **Contextual diversity** has a moderate negative correlation with $\text{cos}_{\text{AVG}}$, suggesting that type-level models may inadvertently

capture topical differences. This has important implications for interpreting cosine-based change estimates, as also noted by Del Tredici et al. (2019).

## 4.3 Qualitative analysis

We now turn to two types of qualitative information: synonyms for the semantic shifts and spontaneous remarks on their use. As previously noted (§ 3.4), in nearly all cases a large majority of participants provided synonyms in line with the posited interpretation of semantic shifts. This is an important general validation of the pipeline used to identify target words and single contact-related examples. As for the spontaneous remarks, key trends for sample items are shown in Table 2; see Appendix C for all semantic shifts and sample verbatim comments. Since the remarks were optional, their presence – even in low absolute numbers – is indicative of salient sociolinguistic properties.

| sem. shift | human$_{\text{AVG}}$ | synonym | loc. | Fr. |
|---|---|---|---|---|
| terrace | $5.67 \pm 0.90$ | patio (7) | 3 | 1 |
| boutique | $5.13 \pm 0.99$ | store (10) | 3 | 3 |
| entourage | $3.20 \pm 1.26$ | friends (5) | 0 | 4 |
| definitively | $2.09 \pm 1.51$ | definitely (11) | 0 | 0 |

Table 2: Examples with acceptability ratings (mean and standard deviation); top synonym with number of speakers who provided it; number of speakers describing the item as locally specific (*loc.*) or linked to French (*Fr.*).

Semantic shifts with high acceptability are generally described as regionally specific (e.g. *terrace* 'restaurant patio'), sometimes together with a perceived role of French (e.g. *boutique* 'shop'). By contrast, words with lower acceptability give rise to more varied comments. Some are more strongly related to a perceived impact of French, such as *entourage* 'close friends'. Most interviewees claim that they would not use it in English, but three indicate that they would in French. As for *definitively* 'for sure', most participants outright reject it as unnatural without further qualification; only one suggests that they use it, and a further two that they might hear it used. But both words are widely attested with contact-related meanings in the Twitter corpus, highlighting the need to nuance the conclusions drawn from this type of data. Moreover, these two words have similar type-level and token-level semantic change scores, but are perceived very differently. This highlights the need to complement quantitative scores with qualitative information.

## 5 Conclusion

We used different embedding-based models to analyze semantic shifts in Quebec English, and assessed them based on acceptability ratings and qualitative remarks by speakers from Montreal. We found that the models were overall relevant, but that (i) model-derived scores aligned differently, and never decisively, with human ratings; (ii) empirical linguistic properties affected all scores, but in different ways; (iii) semantic shifts with similar quantitative scores were often linked to divergent qualitative assessments. This both demonstrates the complementarity of different semantic change estimates and calls for caution in interpreting them, as they fail to capture the full range of information relevant to speakers from the target community.

## Limitations

In the computational experiments, we relied on monolingual models to analyze cross-linguistic phenomena. Our high-level methodological aim – identifying effects of language contact in monolingual data from demographically different speech communities – is grounded in the sociolinguistic literature (Tagliamonte, 2002), but the use of multilingual models is an important alternative to be explored in future work. In the face-to-face interviews, we collected acceptability ratings for a single occurrence of a target word, which limits direct comparability with computational type-level ratings. This was necessary to enable coverage of a relatively large number of target words, and we also took precautions to ensure that the chosen examples were representative (Appendix B). Moreover, due to public health restrictions at the time, we were only able to recruit a participant sample that is comparatively small and is not balanced across sociodemographic characteristics. However, the participants' diversity is reflective of the local sociolinguistic context, and is valuable from a qualitative standpoint. Finally, our analysis focused on a specific type of language variation in a single target speech community. This is a realistic use case scenario for semantic change models which has highlighted important methodological issues, but we are unable to precisely estimate how much these issues would generalize to other applications.

## Ethics Statement

The protocol used to collect and analyze data was approved by the Research Ethics Boards of the Uni-

versité de Toulouse (approval number 2021-396) and of the Université de Sherbrooke (approval number 2022-3289); the latter administered fieldwork-specific funding and facilitated on-site interviews. For further discussion of participant recruitment and informed consent, see Appendix B.

Previously created datasets were used in line with their intended use and licenses. Specifically, the corpus of tweets (§ 3.1) was created, and is distributed for research purposes in the form of tweet IDs,[4] in accordance with Twitter's Developer Agreement and Policy.[5] The binary classification dataset from which we drew the target semantic shifts (§ 3.4) is distributed under the Creative Commons BY-NC-SA 4.0 license.[6] We are releasing the extended set of semantic shifts – containing model-derived semantic change scores, aggregate information from expert and interviewee annotations, and empirical linguistic properties – under the Creative Commons BY-NC-SA 4.0 license.

The results of computational experiments presented here are based on aggregate representations of word meaning; no personally identifiable or otherwise sensitive information is discussed. The individual corpus examples used in face-to-face interviews were manually checked (including for offensiveness) during protocol design, and any personally identifiable information was redacted (Appendix B). The interviewees' sociodemographic characteristics are anonymized and securely stored; we only report aggregate trends. As for more general risks, research on well-defined linguistic communities may be viewed relative to broader public debates, in particular on language policy. We stress that our study is based on an empirical analysis of attested language use and not a predetermined view of Quebec's linguistic communities.

## Acknowledgements

We are grateful to the anonymous reviewers for their remarkably helpful feedback. We are also indebted to our 15 volunteer participants. The research presented here was partly supported by the Fonds de recherche du Québec through a PBEEE short-term research scholarship awarded to the first author. The presented experiments were carried out using the OSIRIM computing platform, administered by the IRIT research laboratory and supported by the CNRS, the Région Occitanie, the French Government and the European Regional Development Fund (see https://osirim.irit.fr).

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

## A  Word embedding models

**Preprocessing.** Before being used in word embedding models, the corpus was tokenized and POS tagged using `twokenize` (Gimpel et al., 2011; Owoputi et al., 2013) and lemmatized using the NLTK WordNet lemmatizer (Bird et al., 2009).

**Type-level embeddings.** We used the best performing type-level configuration from an earlier standard evaluation (Miletić et al., 2021), which was formulated as a binary classification task: `word2vec` skip-gram architecture with negative sampling, 100-dimensional vectors, window size of 5, and minimum word frequency of 100 in each subcorpus. The default values were used for other hyperparameters: negative sampling rate set to 5, subsampling rate set to $10^{-3}$, and number of iterations set to 5. The models were implemented using `gensim` (Řehůřek and Sojka, 2010).

A model was trained independently for each of the three regional subcorpora. In order for their vector spaces to be comparable, the models were aligned using Orthogonal Procrustes (Hamilton et al., 2016); this corresponds to finding the optimal rotational alignment for each pair of matrices. The matrix columns were mean-centered before alignment, as suggested by Schlechtweg et al. (2019).

We used the cosine distance (CD) to estimate semantic change across regional models. In order to control for word embedding instability (Pierrejean and Tanguy, 2018), three models were trained for each subcorpus. We computed the CD for a word $w$ in subcorpora $a$ and $b$ as follows:

$$CD(w_a, w_b) = \frac{\sum_{i=1}^{n} CD(\vec{w}_{a_i}, \vec{w}_{b_i})}{n} \quad (1)$$

for $n = 3$ runs of the model, where $\vec{w}_{a_i}$ is the word's vector corresponding to the subcorpus $a$ in the $i^{th}$ run. We then averaged over the Montreal–Toronto and Montreal–Vancouver distances to obtain an estimate of semantic change.

**Token-level embeddings.** As in previous work (Miletić et al., 2021), we used token-level embeddings from the HuggingFace implementation (Wolf et al., 2020) of `bert-base-uncased`, a 12-layer, 768-dimension version of BERT (110 million parameters) pretrained on English data. No fine-tuning was performed under the assumption that the pretrained model should capture contextual differences necessary to distinguish word meanings.

For each target word, we extracted its occurrences from all three subcorpora; we retained no more than 1,000, and used a random sample for more frequent items. We fed each tweet as a single sequence into BERT, extracted the representations for the target word, and averaged over the last 4 hidden states to get a single contextualized embedding. If a target word was split into subwords by BERT's tokenizer, we averaged over them to get a single embedding.

We identified a word's similar uses by clustering its contextualized embeddings using affinity propagation, in the `scikit-learn` implementation (Pedregosa et al., 2011) with default parameters. Clusters containing at least 5 tweets, more than half of which were published in Montreal, were retained for annotation by domain experts. This was limited to a maximum of 10 clusters per item, starting with those with the highest proportion of tweets from Montreal.

Annotation consisted in cluster-level binary labels (presence vs. absence of contact-related use). A cluster was annotated as contact-related if, in most occurrences in it, the target word's use was regionally specific to Montreal and potentially explained by the influence of French. This was established based on the sociolinguistic literature and lexicographic evidence. We specifically did not consider that contact-related influence was constituted by locally specific proper names, French homographs, structural patterns (e.g. target word in tweet-initial position), or clusters where no reliable determination could be made. A 15-word sample was annotated by two annotators in order to test the reliability of the general procedure, obtaining a reasonably high Cohen's kappa coefficient of 0.55.

**Computational infrastructure.** The experiments were run on a CPU computational cluster, whose one node consists of 2×12 3GHz cores and 192GB RAM. We used 9 `word2vec` models (3 subcorpora × 3 runs); each was trained on 12 cores in ≈ 2 hours. We used pretrained BERT to model and cluster the occurrences of 40 target words; this was completed in ≈ 1 hour on a single node.

## B  Interview protocol

**Interview structure.** We used the protocol developed in the PAC research program (Phonology of Contemporary English: usage, varieties and structure; Przewozny et al., 2020). It is designed to provide good coverage of key linguistic features and sociodemographic properties in different survey locations. It consists of the following tasks:

(i) reading two word lists targeting pronunciation of specific phonemes; (ii) reading a text targeting pronunciation in connected speech; (iii) a conversation with the fieldworker eliciting linguistic and sociodemographic background; (iv) a conversation with another interlocutor (optional in our case).

These tasks were followed by a semantic perception test, which elicited a range of information on 40 semantic shifts attested in tweets from our corpus (§ 3.4). Key among those were acceptability ratings on a scale from 1 to 6, for which the participants were instructed to (i) interpret 1 as "very unnatural, awkward, you would never say something like that", and 6 as "completely natural, just like something you might say"; (ii) rate the word as used in the example, rather than the example as a whole; (iii) provide the rating with reference to their own use of the word rather than the way they think it should be used or they hear others use it.

In choosing the tweets to be shown as examples, we started from those tagged as contact-related in the cluster-level analysis (Appendix A). For each word, 3 annotators then chose 3 potential examples each. Reconciliation was used to pick one, based on (i) idiomaticity and (ii) clarity of the context with respect to the target meaning. All personally identifiable information in the tweets, including user handles and hashtags, was redacted.

**Participant recruitment.** The interviews were conducted in Montreal between mid-January and mid-February 2022. The study was advertised through mailing lists of student associations, posters in public places etc. Recruitment materials briefly explained the study and stated that it was open to "all Montrealers aged 18 or over who are able to conduct a conversation in English".

Potential participants were asked to contact the fieldworker by email, who in turn provided an informed consent form. It outlined the general aims of the study, its procedure, risks and benefits, right to withdraw, and conditions of storing and using the collected data. The precise aims were disclosed after the interview. The consent form also explicitly stated that participation would not be remunerated; note that participation was entirely voluntary and required an active decision on the part of potential participants to contact the fieldworker. A total of 15 participants were recruited.

**Data recording and analysis.** Due to public health restrictions at the time, most interviews (14 out of 15) were conducted remotely, using the Zoom video conferencing platform. For the semantic perception task, the participants used an online LimeSurvey interface, with the Zoom call running in the background; this enabled them to provide qualitative remarks on semantic shifts. The mean interview duration was 1 hour and 15 minutes (min = 56 min; max = 1h 37min).

The participants were asked to provide synonyms for the target word as an interpretation check, but not all of them did so systematically. We retained acceptability ratings lacking a synonym if all remaining speakers had provided the same interpretation. Otherwise, we only retained the ratings where participants had provided a synonym consistent with the target meaning.

**Summary of speaker profiles.** We recruited 10 female, 4 male, and 1 non-binary participant. The distribution of other sociodemographic features is summarized in Table 3.

|          | mean | std  | min   | max  |
|----------|------|------|-------|------|
| **age**     | 37.5 | 18.4 | 19    | 70   |
| **RI**      | 5.4  | 2.8  | 1     | 10   |
| **En.**     | 0.83 | 0.19 | 0.43  | 1.00 |
| **Fr.**     | 0.58 | 0.24 | 0.26  | 1.00 |
| **biling.** | 0.25 | 0.38 | -0.57 | 0.70 |

Table 3: Summary of sociodemographic features

In the table, **RI** refers to the Regionality Index (Chambers and Heisler, 1999), which quantifies links with the local community. In our formulation, it ranges from 1, corresponding to participants who were born, raised, and continue to live in the Montreal region; to 10, corresponding to those who live in Montreal, but were born and raised outside of Canada, as were their parents.

We also computed English and French language use scores (**En.** and **Fr.**) using a previously established procedure (Rouaud, 2019). The scores take into account proficiency, age and manner of acquisition, and frequency and domains of language use; they are normalized to a value range from 0 to 1. We then subtracted the French score from the English score to obtain a bilingualism score (**biling.**). It theoretically ranges from -1 (monolingual French speaker) to 1 (monolingual English speaker).

## C  Qualitative data

This section provides sample verbatim comments for the semantic shifts discussed in Section 4.3; note that italics denote metalinguistic reference and angled brackets denote pronunciation in French. We then provide a summary of synonyms and qualitative tags for all semantic shifts in the test set.

**Sample comments:** *terrace*

Rated example: The weather is still perfect for a lunch on the **terrace** #i♡ny à Greenwich Village

(1) Yeah, pretty normal. Sometimes I'll hear people say *patio* as an alternative, but usually *terrace*.

(2) That sounds super normal for me and very comfortable, and I fully recognize that that's a Montreal thing, or a Quebec thing. You would say, God I don't even remember what the, the non-Montreal version of that sent– that word is, it's not *patio*, it's something like *patio*.

(3) Yeah, exactly what I would use here. *Terrace*, yeah, outdoor, I can't even think of any other word for an outdoor space, it definitely would be *terrace*, like I'm, I'm not sure, but *terrace* is a hundred percent what I would use here.

**Sample comments:** *boutique*

Rated example: The first H&M Home **boutique** will open shortly in Carrefour Laval.

(1) So this is, sounds very normal to me. I guess it would be like, you would say *store*, but *boutique* is just totally normal for me at this point, so that's very not awkward.

(2) Yeah, for sure, we use *boutique* all the time.

(3) I would replace it with *store*, again because I try to avoid French words in, in English, even though English is heavily influenced by French.

**Sample comments:** *entourage*

Rated example: I believe this. I'm not there yet but my **entourage** is all approaching or in their early 30s and I can see/feel how they have a much stronger sense of purpose, direction and self. Gives me something to look forward to and I feel blessed to have people like this around me.

(1) I would say it's a bit unnatural, I would say, for me, I would say *surroundings* instead of *<entourage>*. I would say, I would use *<entourage>* if I was speaking French, yeah.

(2) I would not say *<entourage>* in English speech. I would say *my close friends*, I would try to be more specific I think, *close friends* or *inner circle*, yeah.

(3) Yeah, this makes sense to me, but I wouldn't really use the word *entourage*. I think I'd just use the word *group* maybe, I think the word *group* is all-encompassing in this context and I would say that.

**Sample comments:** *definitively*

Rated example: Thank you! My first try was tasty but kind of "liquidish". I will **definitively** try your recipes.

(1) I feel like somebody must have said it because it doesn't sound weird, I don't know, *defi– I'll definitely*, what, *definitively*, yeah I think I'll leave it, I'll put it at three maybe because it does sound unnatural but it's not maybe that bad. Yeah I guess I would just say *definitely* instead but, yeah.

(2) Yes, people probably say it. *Definitely* but not *definitively*, that's not even what *definitive* means.

(3) I almost, personally for me that's grammatically incorrect, I would say *definitely try your recipes*.

**Summary of collected information**

Table 4 outlines the information collected in the semantic perception task for the full set of 40 semantic shifts.

| | acceptability | | | synonyms | | | social | |
|---|---|---|---|---|---|---|---|---|
| item | mean | std | top 3 synonyms | type | tok. | ttr | loc. | Fr. |
| affirmation | 3.27 | 1.79 | statement (4); claim (2); argument (1) | 7 | 11 | 0.64 | 1 | — |
| ambiance | 4.00 | 1.69 | atmosphere (5); environment (2); vibe (2) | 6 | 12 | 0.50 | — | 2 |
| animator | 4.11 | 1.54 | advisor (2); coordinator (2); counsellor (2) | 10 | 15 | 0.67 | — | 3 |
| availability [pl.] | 5.20 | 1.21 | availability [sg.] (2); when you're free (2); available time (1) | 5 | 7 | 0.71 | 2 | — |
| boutique | 5.13 | 0.99 | store (10); outlet (1); shop (1) | 3 | 12 | 0.25 | 3 | 3 |
| chalet | 4.33 | 2.22 | cottage (6); country (3); cabin (2) | 5 | 14 | 0.36 | 3 | 1 |
| circulation | 3.80 | 1.82 | traffic (11) | 1 | 11 | 0.09 | — | 1 |
| coordinate | 2.70 | 1.64 | contact information (5); information (3); phone number (2) | 8 | 15 | 0.53 | — | 1 |
| deceive | 2.70 | 1.77 | disappoint (6); let down (5) | 2 | 11 | 0.18 | — | — |
| deception | 2.00 | 1.00 | disappointment (2); be disappointed (1) | 2 | 3 | 0.67 | — | — |
| definitively | 2.09 | 1.51 | definitely (11); for sure (1) | 2 | 12 | 0.17 | — | — |
| deputy | 4.20 | 1.94 | elected official (4); MP (2); MNA (1) | 8 | 12 | 0.67 | — | — |
| dossier | 3.62 | 1.77 | issue (3); case (2); enjeu [Fr.] (1) | 9 | 12 | 0.75 | — | 2 |
| entourage | 3.20 | 1.26 | friend (5); surrounding (2); circle (1) | 12 | 17 | 0.71 | — | 4 |
| exchange | 3.27 | 1.98 | talk (6); chat (3); interact (2) | 9 | 18 | 0.50 | — | — |
| exploration | 4.60 | 1.76 | investigation (2); search (2); examination (1) | 7 | 9 | 0.78 | — | — |
| exposition | 4.87 | 1.68 | exhibition (4); show (3); vernissage (2) | 5 | 11 | 0.45 | — | — |
| formation | 2.80 | 1.78 | training (8); formation [Fr.] (1); initiation (1) | 6 | 13 | 0.46 | 1 | 5 |
| formidable | 3.47 | 1.96 | great (6); amazing (2); awesome (1) | 8 | 14 | 0.57 | — | — |
| grave | 1.87 | 1.12 | serious (6); bad (4); grave [Fr.] (1) | 3 | 11 | 0.27 | — | 3 |
| hesitate | 5.27 | 1.16 | debate (3); be hesitant (1); can't decide (1) | 8 | 10 | 0.80 | — | — |
| laureate | 2.67 | 2.08 | winner (2); accomplished people (1) | 2 | 3 | 0.67 | — | — |
| local | 3.50 | 2.12 | arcade (1); neighborhood establishment (1) | 2 | 2 | 1.00 | — | 1 |
| manifestation | 3.67 | 1.88 | protest (6); gathering (2); demonstration (1) | 7 | 13 | 0.54 | — | 6 |
| merit | 3.33 | 1.54 | deserve (11); earn (2) | 2 | 13 | 0.15 | — | — |
| militant | 4.87 | 1.19 | <omission> (2); activist (1); advocate (1) | 7 | 8 | 0.88 | 1 | 2 |
| nomination | 4.53 | 1.96 | appointment (2); <omission> (1); election (1) | 4 | 5 | 0.80 | — | — |
| occasion | 4.47 | 1.55 | opportunity (6); moment (2); time (2) | 4 | 11 | 0.36 | — | 1 |
| pass | 5.53 | 0.92 | come by (4); stop by (4); drop by (3) | 4 | 12 | 0.33 | 1 | — |
| permit | 4.27 | 1.87 | licence (11) | 1 | 11 | 0.09 | 3 | — |
| population | 5.60 | 0.74 | people (6); citizen (1); large group of people (1) | 4 | 9 | 0.44 | — | — |
| portable | 2.87 | 1.92 | phone (6); laptop (3); cell phone (2) | 9 | 19 | 0.47 | — | 2 |
| proposition | 3.33 | 1.80 | suggestion (5); idea (3); proposal (3) | 6 | 14 | 0.43 | — | 2 |
| prudent | 3.80 | 1.74 | careful (7); cautious (4); aware (1) | 5 | 14 | 0.36 | — | — |
| remark | 2.47 | 1.81 | notice (9); realize (1) | 2 | 10 | 0.20 | — | 1 |
| reparation | 2.87 | 1.85 | repair (6); fixing (2); fix (1) | 5 | 11 | 0.45 | — | 2 |
| resume | 2.07 | 1.16 | show (6); sum up (2); summarize (2) | 8 | 15 | 0.53 | — | 4 |
| souvenir | 2.67 | 1.80 | memory (9) | 1 | 9 | 0.11 | 1 | 2 |
| terrace | 5.67 | 0.90 | patio (7); outdoor place (1); outdoor space (1) | 3 | 9 | 0.33 | 3 | 1 |
| trio | 4.80 | 1.74 | combo (6); meal (1); triple order (1) | 3 | 8 | 0.38 | 2 | — |

Table 4: Summary of information collected in the semantic perception test: acceptability ratings (mean and standard deviation); top 3 synonyms with number of speakers who provided them; quantitative characterization of the synonyms (number of types, number of tokens, and type-token ratio); number of qualitative remarks expressing target social values: local specificity (*loc.*) and influence of French (*Fr.*).