# OpenReview forum: "Understanding Computational Models of Semantic Change: New Insights from the Speech Community"
_EMNLP/2023/Conference — EMNLP 2023 Main_

### Official Review · Reviewer_TUJP · 2023-07-28

**Soundness:** 4

**Excitement:**

4: Strong: This paper deepens the understanding of some phenomenon or lowers the barriers to an existing research direction.

**Paper Topic And Main Contributions:**

The paper presents a study of regional semantic variation between English with high French influence in Montreal and English with less French influence (Vancouver and Toronto). The study compares insights obtained from static (type-based) and contextualized (token-based) models for measuring semantic shifts to insights obtained from sociolinguistic interviews with speakers of Montreal English. All experiments are based on 40 known semantic shifts and Twitter corpora from the three different regions. The models and interviews assess different aspects of semantic shifts: static embeddings are used to measure the distance between Montreal English (i.e. high French influence) to Vancouver and Toronto English; contextualized embeddings are used to measure the proportion of contact-related uses (tweets) and the degree of regional specificity via clustering approaches; the interviews provide acceptability ratings of the Montreal variants of the words (and other sociolinguistic information). The paper presents an analysis of how these different aspects interact. While some computational measures reflect human judgments, others do not. A qualitative analysis shows that similar computational scores can occur with highly different human assessments of a shift.

I think the paper presents a valuable example of a sociolinguistically informed study of language models. However, I am not convinced by the argumentation. In particular, I find it very difficult to understand how the validity of the models is assessed (see reasons to reject). It is possible that I misunderstood something and I am willing to change my score if my questions can be clarified in the discussion period.

**Questions For The Authors:**

What exactly are the assumptions behind the different computational measures? If the models give an accurate reflection of the semantic shifts, what should they show? How should the model scores interact with the human acceptability ratings?

**Reasons To Accept:**

The paper presents a (to the best of my knowledge) underrepresented application of language models to synchronic variation (rather than diachronic change.

The paper is well-grounded in work on regional variation and presents a carefully designed and collected dataset of speaker judgments. As such, it serves as a good example of computational work grounded in sociolinguistic research and data.

The study aims to control for several factors that may impact the reflection of shifts (e.g. frequency and polysemy).

**Reasons To Reject:**

Core research questions and motivation of analysis: Ultimately, I found it very difficult to follow the core reasoning behind the experiments and observations. What should the two models show if they can reflect regional shifts accurately? What would a valid result look like?

Methods for measuring change using language models: I was left wondering what exactly the two different ways of querying models should reflect and how they relate to each other. If I understand correctly, the method uses clustering (in combination with human labeling of clusters) to identify the tweets in which words are used in the language-contact sense (i.e. English words with high French influence used in English tweets). Two scores are calculated using this information: the proportion of contact-related tweets and the proportion of tweets from Montreal out of all contact-related tweets. The first score is argued to reflect the defusion of the contact-related word and the second score the regional specificity. The static method simply reflects the cosine difference between tweets from Montreal (high contact with French) to tweets from Vancouver and Toronto (low contact). To me, the intuition behind what the methods should show if they accurately reflect change is not completely clear.

Validity analysis and interference between genre and region: The dataset only contains words that are affected by variation due to language contact. The scores obtained from the models are compared to human acceptability ratings of the target words. I am wondering whether factors not taken into account in the study could impact the comparison. Specifically, I could imagine that a twitter corpus may reflect word usages that are specific to online discourse (in particular Twitter discourse) that might not be seen as acceptable by all speakers.

Update after rebuttal: Thank you for the clarifications! The explanations helped a lot and I am confident that the authors can make the necessary changes to their final version. I am thus increasing my soundness score.

**Reproducibility:**

4: Could mostly reproduce the results, but there may be some variation because of sample variance or minor variations in their interpretation of the protocol or method.

**Reviewer Confidence:**

4: Quite sure. I tried to check the important points carefully. It's unlikely, though conceivable, that I missed something that should affect my ratings.

---

> ### Author Rebuttal · Authors · 2023-08-28
>
> Thank you for your thorough and thought-provoking review, as well as for highlighting the relevance of our synchronic and sociolinguistic perspective.
>
> **Assumptions behind semantic change estimates:** In assessing several estimates, we aimed to account for different types of information that can be derived across modeling strategies. While the patterns they capture may partly overlap, our core assumptions are as follows.
> * *cos-avg* reflects the assumption that a contact-induced semantic shift should have a different static word embedding in the Montreal data compared to Toronto and Vancouver. This score should be higher for clear-cut usage differences (e.g. *definitively* ‘for sure’) than for those that are finer-grained (e.g. *population* ‘the people’).
> * *clust-cont* indicates the proportion of tweets in which a target word is used with a contact meaning. Since the vast majority of tweets containing a target word come from different users, we assume that this score indicates the diffusion of the contact-related use. It should correlate with estimates of diffusion measured in face-to-face communication.
> * *clust-reg* indicates the proportion of contact-related uses of a target word posted in Montreal, i.e. it measures their local specificity. We assume that this property varies for different reasons, e.g. an emergent semantic shift may be specific to Montreal, but an established one may spread elsewhere.
>
> Thank you for pointing out the lack of clarity on this key issue. We will include more extensive explanations.
>
> **Establishing validity using human acceptability ratings:** In strict terms, we would consider that a computational method is validated by human acceptability ratings if it closely aligns with them. However, our motivation was broader: we wanted to gain insights into a range of previously proposed computational semantic change estimates by examining how they relate to a standard sociolinguistic estimate of the same phenomenon. Our main aim was therefore to understand how these estimates pattern with one another rather than establish a strict validity threshold. In retrospect, we realize that framing our analysis in terms of validity may have obscured this perspective. We will revise our phrasing in order to clarify this.
>
> **Confounding effect of genre (Twitter discourse):** We aimed to control for the potential effect of online-specific usage in several ways.
> * The rated examples were selected by three expert annotators, who prioritized tweets which did not read as stylistically marked.
> * We collected background information about the informants' perception of language use on social media.
> * We also instructed them to rate acceptability based on the usage of the target word rather than properties of its context.
>
> Some of these points are explained in Appendix B, but we will try to include more details in the body of the paper as well as acknowledge the remaining potential effect of genre variation.

---

### Official Review · Reviewer_a31m · 2023-07-31

**Soundness:** 3

**Excitement:**

3: Ambivalent: It has merits (e.g., it reports state-of-the-art results, the idea is nice), but there are key weaknesses (e.g., it describes incremental work), and it can significantly benefit from another round of revision. However, I won't object to accepting it if my co-reviewers champion it.

**Paper Topic And Main Contributions:**

This paper investigates how the output of lexical semantic change detection systems is perceived by the speakers of a given community. The focus is on Quebec English and the semantic shift that geographic contact with French may induce. The authors use an existing corpus of Canadian English tweets, focusing on language use specific to Montreal, where both English and French are widely used. They consider 40 words which previous work had identified as undergoing semantic shift due to language contact reasons. They derive type- and token-based word embeddings for these words and compute cosine distance across regions. The main contribution of the paper is to evaluate this computational estimates of semantic shift via face-to-face sociolinguistic interviews with 15 speakers from the community under study.

**Questions For The Authors:**

- Question A: could you please clarify in what way acceptability ratings reflect the local diffusion of contact-related uses? (Line 194)

- Question B: If acceptability is assumed to be related to diffusion, what is the motivation behind testing whether it correlates with semantic distance (i.e. degree of shift)? I don't quite understand how acceptability ratings may help us to evaluate the cosine distance scores (the examples in 223-232 didn't help me unfortunately). Unlike in the study by Del Tredici et al. (2019), the speakers in this case did not evaluate degree of shift between two usages -- instead, they were asked to judge the acceptability of single usages (one single tweet per word).

- Questions C: Are the correlations reported in sections 4.1 and 4.2 statistically significant? Please indicate this in the paper.

**Reasons To Accept:**

- Interesting use of synchronic data from language contact regions to investigate semantic shift (rather than using diachronic data).
- Qualitative analysis to complement and validate insights from authomatic methods for seamntic shift detection

**Reasons To Reject:**

- While the use of qualitative analysis with community members is a strength, the current study has a very small scale: just 15 participants from only Montreal. This leads to not very robust claims (e.g., claiming at lines 287-288 that "semantic shifts with high acceptability are generally described as regionally specific" when only 3 out 15 participants have descrbed a word as locally specific, according to Table 2).

- Using written text (tweets) for this kind of study has the potential of confounding semantic shift with code-switching in cases where this would only be noticeable in spoken language. Lines 234-235 hint at this problem. While this may not be prominent in the current data, it seems a fundamental weakness of the approach.

**Reproducibility:**

4: Could mostly reproduce the results, but there may be some variation because of sample variance or minor variations in their interpretation of the protocol or method.

**Reviewer Confidence:**

5: Positive that my evaluation is correct. I read the paper very carefully and I am very familiar with related work.

**Typos Grammar Style And Presentation Improvements:**

It would be useful to provide one or two examples of the 40 words considered in section 3.2 and explain more explicitely what is meant by "contact meanings" (line 126).

---

> ### Author Rebuttal · Authors · 2023-08-28
>
> Thank you for your detailed comments and for underscoring the relevance of our synchronic analysis.
>
> **“A bit thin” amount of substance in the paper:** We respectfully disagree with this characterization. This short paper does not aim to propose a new computational method. Instead, we introduce a carefully designed and collected dataset, and use it to analyze existing computational methods. This analysis is strongly grounded in sociolinguistic practice and thereby provides a novel perspective on semantic change models, which we believe is a valuable contribution in its own right.
>
> **Small scale of the study:** We are aware of suboptimal aspects of our 15-participant sample (as discussed in Limitations), but we would not say that this is “a very small scale by any standards”. We deliberately collected in-depth qualitative information, for which we had to compensate by limiting the number of participants and rated instances; however, all participants rated all of the instances. We think this is a reasonable trade-off to obtain qualitative data, which is underrepresented in analyses of semantic change models. Regarding the robustness of claims, participants’ comments on local specificity and French influence were optional; a low absolute number may be relevant in relative terms, but we will clarify the strength of our claims.
>
> **Semantic shifts vs. code-switching in tweets:** As pointed out, we noted this issue for one example (l. 231-235), but it is overall rare in our data. We consider this as a minor tradeoff for access to orders of magnitude more data compared e.g. to interview recordings. We are unaware of alternatives which would allow us to model contact-induced semantic shifts at scale without this potential confound.
>
> **Question A:** We expect acceptability ratings to directly reflect the diffusion of semantic shifts among our participants, since they were instructed to provide the ratings based on their own language use.
> A more general assumption is that items which are more widespread in the broader speech community give rise to higher acceptability ratings. We accept that this link is indirect, and will clarify our assumptions.
>
> **Question B:** While we think that acceptability primarily reflects diffusion, it may also interact with other estimates, including the extent of change measured by cosine distance. We agree that this is not a targeted evaluation such as that of Del Tredici et al. (2019); our aim was to gain insights into how a standard sociolinguistic measure such as acceptability relates to a range of previously proposed computational estimates of semantic change. We will clarify this.
>
> **Question C:** In Section 4.1, the correlation with *cos-avg* is statistically significant, while those with *clust-cont* and *clust-reg* are not. Thank you for spotting this issue; we will report significance values. In Section 4.2, statistical significance is indicated through shading in Table 1, and the subsequent discussion is limited to statistically significant correlations (as noted l. 251).
>
> **Presentation:** Thank you for your suggestions; will provide examples and more detailed definitions in Section 3.2.

---

### Official Review · Reviewer_3VVP · 2023-08-04

**Soundness:** 3

**Excitement:**

4: Strong: This paper deepens the understanding of some phenomenon or lowers the barriers to an existing research direction.

**Paper Topic And Main Contributions:**

The paper employs computational methodologies to study semantic change to study contact-induced semantic shifts in Quebec English. Not only they aim to provide insights into the phenomenon, but they investigate the validity of these methods for a sociolinguistic study on speech, where they have not been extensively applied yet (the authors claim to be the first). The results indicate the applicability of the method overall but with some limits and caveats to consider.

**Questions For The Authors:**

1. Do you think that the fact that you used a model trained on "generic" English (not associated with a specific country/region) could have some effect on the results?
2. It seems to me that it could be possible to obtain further insights into the contact-induced semantic shifts, by comparing the location-specific embeddings also with corresponding French word embeddings. Do you have any ideas for future work in this direction?

**Reasons To Accept:**

The paper is well written and reports on a thorough analyses both in the computational modeling and in the elicitation of judgements. The authors manage to report their work in detail, in spite of the 4-page limit.

The authors well describe the results of their study without simplifications and with attention to potential confounders.

**Reasons To Reject:**

Some of the results on the correlation between the computational scores and the human judgments remain inconclusive. Consequentially, the applicability of the methods is not fully validated. It is not clear if this is an issue of the general methodology often employed in the field, or of specifics of the way it is instantiated in the current study.

Acceptability judgments elicited in interviews may not be as flexible as in the context of online speech. This may be a factor in the lack of a strong alignment with the semantic shift scores computed from Twitter data.

**Reproducibility:**

4: Could mostly reproduce the results, but there may be some variation because of sample variance or minor variations in their interpretation of the protocol or method.

**Reviewer Confidence:**

4: Quite sure. I tried to check the important points carefully. It's unlikely, though conceivable, that I missed something that should affect my ratings.

---

> ### Author Rebuttal · Authors · 2023-08-28
>
> Thank you for your feedback and for pointing out the novelty of our sociolinguistic approach.
>
> **Incomplete validation of the methods:** Our aim was not to optimize the validity of a specific method, but to assess existing approaches based on how their output is perceived by target speakers; we will clarify this. Low correlations between human judgments and some model-derived scores are informative in this respect.
>
> **Acceptability ratings in online communication vs. interviews:** We accept that there may be differences between the two settings, but this is inherent in our aim of assessing the outputs of Twitter-based models. We took care to explain the origin of the examples to the participants. We also adopted standard sociolinguistic practice to decrease the degree of formality, crucially by conducting the rating task at the end of the interview, when participants generally communicate more freely.
>
> **Question 1:** We deliberately chose BERT pretrained on generic English data so as to assess a widely used standard method. This may have some effect on a subset of our results, but we expect it to be minor. We assume that the model still captures contextual differences useful in disambiguating target word uses. We model and cluster the occurrences of one target word at a time, which should limit potential discrepancies in representations of different target words. A direct comparison with a fine-tuned model is a relevant direction of future work.
>
> **Question 2:** We believe that using French (or multilingual) word embedding models is a good idea which would enable us to directly compare the meanings of candidate semantic shifts in English and their French equivalents. As noted in Limitations, this is a promising methodological alternative, but it falls outside of the scope of this paper.

---

### Meta-Review · Area_Chair_mGvn · 2023-09-10

**Recommendation:** 5

**Metareview:**

The reviewers agree that the paper is very interesting in that it uses diachronic semantic change models to investigate synchronic variation (in addition to presenting a new dataset).
Qualitative analysis is persuasive and the work is extremely well grounded in sociolinguistics. It is even more impressive for a short paper.

The questions posed by the reviewers got persuasive answers from the authors in the rebuttal phase. The authors should make sure to include the main points of these answers in the camera-ready version of the paper.

---

### Decision · Program_Chairs · 2023-10-07

**Decision:**

Accept-Main

**Comment:**

The reviewers agree that the paper is very interesting in that it uses diachronic semantic change models to investigate synchronic variation (in addition to presenting a new dataset).
Qualitative analysis is persuasive and the work is extremely well grounded in sociolinguistics. It is even more impressive for a short paper.

The questions posed by the reviewers got persuasive answers from the authors in the rebuttal phase. The authors should make sure to include the main points of these answers in the camera-ready version of the paper.